# OPPO: ACCELERATING PPO-BASED RLHF VIA PIPELINE OVERLAP

**Kaizhuo Yan[1*]**   **Yingjie Yu[1*]**   **Yifan Yu[1]**   **Haizhong Zheng[2]**   **Fan Lai[1]**

[1]University of Illinois Urbana–Champaign   [2]Carnegie Mellon University

{kaizhuo2, yyu69, yifanyu4, fanlai}@illinois.edu
haizhonz@andrew.cmu.edu        *Equal contribution

## ABSTRACT

Proximal Policy Optimization (PPO)-based reinforcement learning from human feedback (RLHF) is a widely adopted paradigm for aligning large language models (LLMs) with human preferences. However, its training pipeline suffers from substantial inefficiencies due to sequential multi-model dependencies (e.g., reward model depends on actor outputs) and long-tail response lengths, where a few long responses straggle the stage completion. We present OPPO, a novel, lightweight, and model-agnostic PPO-based RLHF framework that improves training efficiency by overlapping pipeline execution. OPPO introduces two novel techniques: (1) Intra-step overlap, which streams upstream model outputs (e.g., actor model) in right-sized chunks, enabling the downstream model (e.g., reward) to begin prefill while the upstream continues decoding; and (2) Inter-step overlap, which adaptively overcommits a few prompts and defers long generations to future steps, mitigating tail latency without discarding partial work. OPPO integrates easily with existing PPO implementations with a lightweight wrapper. Extensive evaluations show that OPPO accelerates PPO-based RLHF training by $1.8\times$–$2.8\times$ and improves GPU utilization by $1.4\times$–$2.1\times$ without compromising training convergence.

## 1 INTRODUCTION

Reinforcement Learning from Human Feedback (RLHF) has become a cornerstone for aligning large language models (LLMs) with human preferences. Among RLHF methods, Proximal Policy Optimization (PPO) (Schulman et al., 2017) has been the de facto standard due to its training stability and flexibility across diverse reward models and objectives. Following InstructGPT (Ouyang et al., 2022), PPO remains the standard for online alignment in both research and industry. Recent work shows it outperforms offline methods like DPO on reasoning tasks (Xu et al., 2024), and it supports massive-scale training in modern tool chains (Shen et al., 2024). A standard PPO-based RLHF pipeline involves four models: an *actor* (policy), a *critic* (value function), a *reference policy* (for KL regularization), and a *reward model* trained on human-labeled preferences. Each training step consists of three sequential stages: (1) *Generation*: the actor generates responses to prompts; (2) *Scoring*: responses are evaluated by the critic, reference, and reward models; and (3) *Training*: the actor and critic models are updated using advantage estimates and gradients.

Despite its effectiveness, PPO-based RLHF faces significant training inefficiencies rooted in its multi-model dependencies. Running and coordinating four LLMs imposes substantial resource requirements, and each stage is constrained by its slowest component. For example, the actor model's generation suffers from severe long-tail latency: a few long responses can delay downstream stages, such as the reward and value models, leading to idle resources and poor training throughput of the pipeline. As LLMs grow larger and context lengths increase, these bottlenecks worsen, (Grattafiori et al., 2024) making PPO-based RLHF increasingly costly to train (§2.2).

Recent advances tackle PPO-based RLHF inefficiencies from both algorithmic and system perspectives. On the algorithmic side, methods such as Direct Preference Optimization (DPO) (Rafailov et al., 2024) and Group-Relative Policy Optimization (GRPO) (Shao et al., 2024) remove components like the value or reward model. However, these approaches often suffer from instability due to sparse rewards, requiring many rollouts to capture intrinsic advantages, and face task-specific

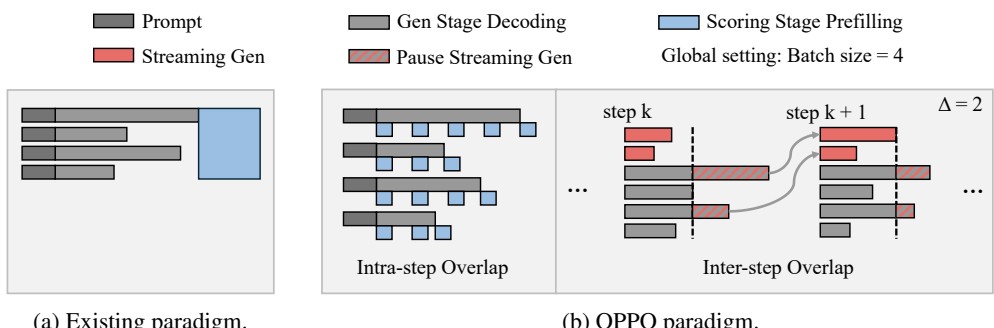

(a) Existing paradigm.  (b) OPPO paradigm.

Figure 1: (a) In the existing paradigm, the scoring stage begins only after the response is fully generated. In contrast, (b) the OPPO paradigm interleaves scoring with generation without altering the final responses (intra-step overlap), and carries unfinished overcommitted sequences into the next iteration (*inter-step overlap*). A batch size of 4 and an overcommitment degree of 2 in illustrations.

reward design challenges (Feng et al., 2024; Fisch et al., 2025; Chen et al., 2024). Asynchronous RLHF (Noukhovitch et al., 2025) reduces pipeline dependencies but introduces staleness, which can harm convergence (§2.2). In contrast, system-level approaches, such as RLHFuse (Zhong et al., 2025), AReal (Fu et al., 2025), and Verl (Sheng et al., 2025), improve throughput via fine-grained parallelism, for example by collocating models to reduce communication overhead or dynamically scaling GPU resources to match workload demands.

In this paper, we explore a complementary opportunity to accelerate PPO-based RLHF: *maximizing execution overlap in the training pipeline*. We introduce two novel insights: (1) *Intra-step Overlap*, which streams tokens from the actor to downstream models, enabling generation and scoring stages to overlap the execution without altering the generated response; and (2) *Inter-step overlap*, which selective overcommits a few prompts per batch and prioritizes faster completions, deferring stragglers to future iterations to hide tail latency without wasting partial generation.

Realizing both overlaps introduces non-trivial challenges. First, overlapping generation and scoring can hide the prefilling latency of downstream models during the decoding execution of the actor model, but also increases resource contention, risking slowing generation due to concurrent executions. Second, excessive overcommitment inflates batch sizes, deferring too many responses per iteration. This not only raises per-batch latency but introduces staleness, ultimately harming convergence.

**Contributions.** In this paper, we present OPPO, a lightweight PPO-based RLHF training framework that improves training efficiency via pipeline overlap, minimizing idle time without compromising convergence. OPPO novelly addresses the aforementioned challenges:

- *Intra-step Overlap*: While the actor generates responses, OPPO streams newly generated tokens to downstream models (e.g., reward model) in adaptive chunks. This enables incremental prefilling and overlaps the generation and scoring stages. Chunk sizes are automatically adjusted online, based on the leftover resources, to balance overlap against resource contention, preserving algorithm correctness and stability (§3.1).
- *Inter-step Overlap*: To mitigate long-tail latency, OPPO adaptively overcommits a few prompts per step. Long-response generations are deferred and resumed in future iterations, preserving partial work and maintaining batch size. It adapts the overcommitment level online, trading small statistical deviations (e.g., reward differences) for large throughput gains (§3.2).
- *Generalized and lightweight*: Our evaluations show that OPPO achieves $1.8\times$–$2.8\times$ speedup and improves GPU utilization by $1.4\times$–$2.1\times$ for PPO-based RLHF with only a lightweight wrapper, and generalizes to other paradigms such as DPO with similar benefits (§4).

## 2  BACKGROUND AND MOTIVATION

We next outline the PPO-based RLHF framework (§2.1), then highlight key inefficiencies in existing training designs that motivate our work (§2.2).

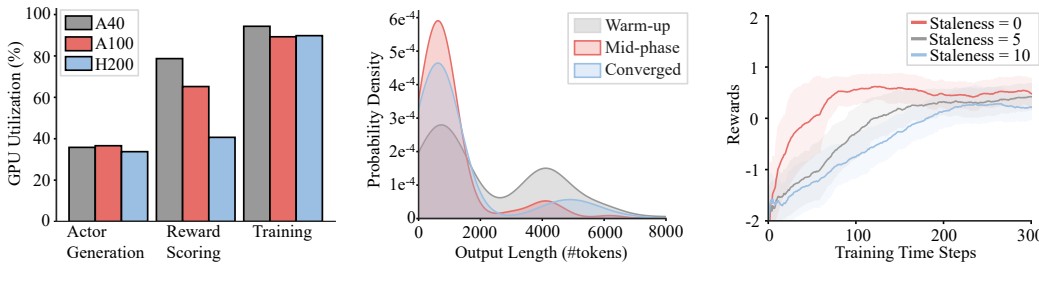

(a) GPU util. varies across stages.  (b) Heterogeneous rollout lengths.  (c) Asynchrony hurts quality.

Figure 2: PPO-based RLHF faces (a) varying resource demands across pipeline stages, and (b) heterogeneous rollout lengths, both of which can produce stragglers that prolong step execution. (c) Existing approaches for asynchronous training risk harming convergence.

## 2.1 BACKGROUND: PPO-BASED RLHF

Figure 1a depicts a single step of a standard PPO-based RLHF pipeline. Given a batch of prompts, the actor model generates output sequences. These are then scored by the reward model, producing scalar rewards that reflect alignment with human preferences. A reference model, typically a frozen copy of the base pretrained model, computes a KL divergence penalty that regularizes the update, discouraging the new policy from drifting too far from the original distribution.

The value model estimates the expected return of each sequence and computes its advantage $\hat{A}_t$:

$$\delta_t = r_t + \gamma V\left(s_{t+1}\right) - V\left(s_t\right), \quad \hat{A}_t = \sum_{\ell=0}^{T-t-1} (\gamma\lambda)^\ell \delta_{t+\ell} \tag{1}$$

where $r_t$ is the reward at step $t$, $V\left(s_t\right)$ is the estimated value of state $s_t$, $\gamma$ is the discount factor, and $\lambda$ is the generalized advantage estimation (GAE) parameter. In every step, actor model is updated by optimizing the clipped surrogate objective:

$$\mathcal{L}^{\text{clip}}(\theta_j) = \mathbb{E}t \sim D\theta_{j-1} \left[\min\left(r_t(\theta_j)\hat{A}t, \text{ clip}\left(r_t(\theta_j), 1-\epsilon, 1+\epsilon\right)\hat{A}t\right)\right],$$

$$\text{where } r_t\left(\theta_j\right) = \frac{\pi_{\theta_j}\left(a_t \mid s_t\right)}{\pi_{\theta_{j-1}}\left(a_t \mid s_t\right)} \tag{2}$$

These four models (actor, reward, reference, and value) form a tightly coupled intra-step pipeline spanning generation, scoring, and training stages. PPO-based RLHF typically runs hundreds of such iterative steps, creating an inter-step pipeline across updates.

## 2.2 TRAINING INEFFICIENCY OF PPO-BASED RLHF

Unlike pre-training a single model, PPO-based RLHF pipelines introduce two types of execution dependencies that limit hardware utilization and training speed: *intra-step* and *inter-step* dependencies.

**Inefficiency due to Intra-step Dependency.** Each model in the RLHF pipeline (e.g., actor, reward, and value models) exhibits distinct computational characteristics. Figure 2a compares GPU utilization across three GPU types (A40, A100, H200). Response generation in the actor model is memory-intensive due to autoregressive per-token decoding, resulting in low GPU utilization (<40%), whereas scoring and training stages are relatively compute-intensive (e.g., due to long-context prefilling in scoring). This heterogeneous resource utilization highlights how mismatched compute demands across stages create idle GPU time, motivating designs to scavenge unused resources.

The inefficiency is further amplified by the long-tailed distribution of response lengths (Figure 2b). While most sequences are short, a subset of responses are significantly longer. Since stage completion

depends on the last sequence (rollout), these heterogeneous lengths introduce tail stragglers. Worse, the length distribution evolves across stages (e.g., at the warm-up stage and converged stage), making optimizations such as dynamically resizing GPU allocations challenging.

**Inefficiency due to Inter-step Dependency.** Each step involves updating model weights. A natural strategy to improve throughput is to tolerate training staleness, where the reward model evaluates actor outputs from previous steps instead of synchronizing in real time, as in AReal (Fu et al., 2025). However, as shown in Figure 2c, asynchronous training (e.g., with staleness 5) can not only slow step-to-reward convergence but also lowers the post-training model quality, emphasizing the need for careful inter-step synchronization in RLHF pipelines.

## 3 OPPO: OVERLAPPING PPO-BASED RLHF TRAINING PIPELINES

To address both intra- and inter-step inefficiencies, we introduce OPPO, an Overlapped PPO-based RLHF training paradigm. As illustrated in Figure 1, OPPO overlaps the training stages to reduce idle time and improve resource efficiency, tackling two key sources of step latency: sequential stage dependencies within a step, and the long-tailed distribution of output lengths. At its core are two complementary techniques: (1) *intra-step overlap*, which overlaps reward scoring with actor generation within a single step, and (2) *inter-step overlap*, which selectively overcommits a few prompts and carries unfinished prompts into the next step to mitigate tail-induced stalling.

### 3.1 OVERLAPPING INTRA-STEP TRAINING PIPELINE

Sequential dependencies across pipeline stages and the long-tailed distribution of response lengths often block downstream execution in PPO-based RLHF. For example, the reward model cannot begin the scoring of a sequence (rollout) until the actor completes generation for that sequence, leading to idle resources and underutilized GPUs. At the same time, heterogeneous resource utilization across models presents a new opportunity: while the upstream actor continues memory-intensive decoding, downstream operators (e.g., reward model) can start the (sub)prefilling of partial outputs in a streaming manner.

By dividing actor generation into chunks and streaming them to the reward model, OPPO overlaps the actor decoding stage with the reward prefilling stage, hiding latency and reducing execution bubbles. This design naturally benefits setups where models are placed on separate GPUs, but also improves efficiency when models are colocated, due to their mismatched compute demands (Figure 2a). To realize intra-step overlap, OPPO partitions actor outputs into right-sized chunks and streams each chunk to the reward model as it is generated. Scoring proceeds progressively within each PPO step: while the actor decodes the $k$-th chunk, the reward model concurrently processes the prefilling of $(k-1)$-th chunk. At the end of the step, the reward model completes prefilling for the final chunk and computes the score based on the entire sequence, whose previous chunks have been processed.

Importantly, this streaming does not alter the response generation $y_i$, the policy log-probabilities, or the critic/value terms used in computing the advantage $\hat{A}(y_i)$. Formally, letting $y_i$ denote the full response and $y_i^{(1)}, \ldots, y_i^{(T_i)}$ its prefixes with $y_i^{(T_i)} = y_i$, the streamed gradient estimator is

$$\widehat{\mathbf{g}}_{\text{str}}(\theta) = \frac{1}{B} \sum_{i=1}^{B} \sum_{t=1}^{T_i} \mathbf{1}_{\text{fin}}^{(i,t)} \hat{A}(y_i) \nabla_\theta \log \pi_\theta(y_i \mid x_i), \tag{3}$$

where $\mathbf{1}_{\text{fin}}^{(i,t)}$ marks the final prefix. Because each sample follows exactly the same prefix, the inner sum collapses, and $\widehat{\mathbf{g}}_{\text{str}}(\theta) \equiv \widehat{\mathbf{g}}_{\text{std}}(\theta)$ point-wise. Thus, intra-step streaming does not change the PPO update, preserving both expectation and variance of the gradient estimator.

**Dynamic Control on Intra-step Overlap.** However, streaming introduces a tradeoff in chunk size. As shown in Figure 7b, large chunks (e.g., 3K tokens) result in low overlap, reducing the benefits of intra-step streaming and reverting to baseline sequential execution. Conversely, small chunks (e.g., 10 tokens) can cause severe resource contention, especially when models are colocated, due to frequent GPU context switching to execute different models. OPPO addresses this by exploiting two key insights: (1) the tradeoff between chunk size and overlap efficiency is monotonic and predictable,

---

**Algorithm 1** OPPO Training with Intra-step and Inter-step Overlap

---

**Require:** Batch size $B$, initial $\Delta$, chunk size $C$, window size $W$, bounds $\Delta_{\min}, \Delta_{\max}$

  1: Initialize Buffer $\leftarrow$ FIFO(capacity $= B + \Delta$); reward_scores $\leftarrow []$
  2: **for** each training iteration **do**                                 ▷ Stage 1: Fill buffer to capacity
  3:     **while** $|\text{Buffer}| < B + \Delta$ **do**
  4:         Buffer.add(sample_from_dataset())
  5:     **end while**                           ▷ Stage 2: Generation with intra-step overlap
  6:     finished $\leftarrow \emptyset$
  7:     **while** $|\text{finished}| < B$ **do**
  8:         active $\leftarrow$ Buffer.get_unfinished()
  9:         **if** $|\text{active}| = 0$ **then**
10:             **break**
11:         **end if**
12:         **parallel do**
13:             chunks $\leftarrow$ Actor.generate_chunk(active, size $= C$)
14:             Reward.reward_incremental(active)      ▷ Finished→prefill+decode; else→prefill.
15:         Update_states(active, chunks)
16:     **end while**                          ▷ Stage 3: PPO update with inter-step overlap
17:     ppo_batch $\leftarrow$ finished$[: B]$
18:     reward_scores.append(ppo_batch.$r$)
19:     PPO.step(ppo_batch)
20:     Buffer.remove(ppo_batch)               ▷ Unfinished sequences remain for next iteration
21:     **if** $|\text{reward\_scores}| \geq 2W$ **then**                   ▷ Dynamic $\Delta$ update
22:         $d \leftarrow$ mean(reward_scores$[-W:]$) $-$ mean(reward_scores$[-2W:-W]$)
23:         $\Delta_{\text{change}} \leftarrow \max\big(1, \lfloor \Delta/4 \rfloor\big)$
24:         $\Delta \leftarrow \text{clip}(\Delta - \text{sign}(d) \cdot \Delta_{\text{change}}, \; \Delta_{\min}, \; \Delta_{\max})$
25:         Buffer.set_capacity($B + \Delta$)
26:         reward_scores = reward_scores$[-W:]$
27:     **end if**
28: **end for**

---

and (2) PPO training runs for many steps, allowing ample opportunities for exploration. Therefore, OPPO periodically (e.g., every 50 training steps) applies a few candidate chunk sizes (e.g., 128, 256, 512) across different steps and selects the best-performing configuration for subsequent windows.

### 3.2 OVERLAPPING INTER-STEP TRAINING PIPELINE

While intra-step overlap improves efficiency within a single PPO step, it does not fully address tail latency caused by the heterogeneous response lengths within a batch. Here, the response must complete generation before reward scoring and subsequent policy updates. Due to the long-tailed distribution of generation lengths, a few slow prompts can delay the entire step. This motivates an inter-step design that allows overlapping across PPO steps without hurting convergence.

OPPO addresses this challenge by overcommitting a few additional prompts per batch to mitigate long-tail stragglers. Specifically, if the original batch size is $B$, OPPO executes $B + \Delta$ prompts per step. The key insight is that sequence generation is typically not computation-bound, so adding a few extra prompts has minimal impact on per-batch execution time while substantially reducing the effect of long-tail sequences. During each step, the first $B$ completed prompts are used for PPO updates, while unfinished $\Delta$ sequences are deferred to the next step. This mechanism ensures that long sequences are not starved, finishing in subsequent steps, and partial work (generation) is preserved across steps.

The overall procedure, combining intra- and inter-step overlap, is summarized in Algorithm 1, where the buffer holds up to $B + \Delta$ sequences, and generation proceeds in parallel with intra-step streaming. The threshold $\Delta$, which controlling the number of unfinished sequences carried over to the next step, introduces a tradeoff between efficiency and convergence. A small $\Delta$ reduces overlap and may leave GPUs idle due to tail sequences, while a large $\Delta$ increases overlap but risks inflating per-step latency and introducing staleness in the PPO update.

**Dynamic Control on Inter-step Overlap.** OPPO automatically adjusts $\Delta$ based on training dynamics. Let $R_t$ denote the average reward in step $t$, and consider a sliding window of $w$ steps. Define the slope of improvement over the window as $s_t = \frac{1}{w} \sum_{i=t-w+1}^{t} (R_i - R_{i-1})$. The threshold $\Delta$ is then updated according to

$$\Delta_{t+1} = \begin{cases} \min(\Delta_{\max}, \Delta_t + \delta_{\text{inc}}) & \text{if } s_t > 0 \,, \\ \max(\Delta_{\min}, \Delta_t - \delta_{\text{dec}}) & \text{if } s_t \leq 0 \,, \end{cases} \quad (4)$$

where $\delta_{\text{inc}}$ and $\delta_{\text{dec}}$ are fixed momentum (e.g., 1), and $\Delta_{\min}$ and $\Delta_{\max}$ are bounds on the buffer size. As training starts to converge and $s_t \to 0$, $\Delta_t$ naturally decays toward $\Delta_{\min}$ (often zero), preventing overcomittement to ensure convergence while effectively mitigating tail-induced delays across steps.

## 4 EVALUATIONS

### 4.1 EXPERIMENTAL SETUP

All experiments are conducted on high-end NVIDIA GPUs with different configurations. Stack-Exchange-Paired with Qwen2.5-7B-Instruct is run on 8×H200 (141GB) GPUs, while GSM8K with Qwen2.5-7B runs on 4×GH200 (96GB) GPUs. Stack-Exchange-Paired with Qwen2.5-3B-Instruct and OpenCoder-SFT with Qwen2.5-3B-Instruct are executed on 8×A100 (80GB) GPUs.

**Models & Datasets.** We follow state-of-the-art PPO settings using the Transformer Reinforcement Learning (TRL) library (von Werra et al., 2020a). For actor models, we experiment with Qwen2.5-7B, Qwen2.5-7B-Instruct, and Qwen2.5-3B-Instruct, each augmented with a value head for PPO optimization. The reward model is either a Qwen2.5-7B or a rule-based evaluator (for math tasks). We evaluate on three popular tasks widely used in RLHF research (detailed evaluation setup in Appendix A.1):

- *Free-form generation*: Stack-Exchange-Paired (von Werra et al., 2020b), which contains QA pairs with preference labels.
- *Math reasoning*: GSM8K (Cobbe et al., 2021), which consists of grade-school math word problems. We convert it into preference format by ranking paired outputs by correctness and reasoning clarity.
- *Code generation*: OpenCoder-SFT (Stage 2) (Huang et al., 2024), which contains large-scale programming tasks across multiple languages.

**Baselines.** We follow the standard distributed PPO setting. Based on the memory and computation resource requirements of each model, we allocate seven GPUs to the generation and training stages, and one GPU to the scoring stage (i.e., reward model). We compare OPPO against TRL's PPO (von Werra et al., 2020a), the state-of-the-art and widely adopted framework in PPO. It is worth noting that OPPO is complementary to existing PPO frameworks and can be integrated into them. Unless otherwise specified, we use a training batch size of 112.

**Metrics.** We evaluate both efficiency and quality. Efficiency is measured by training speed, including time-to-reward and step-to-reward. Quality is measured by the final achieved reward. All results are averaged over five independent runs.

### 4.2 END-TO-END PERFORMANCE COMPARISON

We start by evaluating OPPO's end-to-end efficiency and quality performance.

**OPPO achieves substantial PPO training speedup.** Figure 3 shows that OPPO consistently accelerates PPO training by $1.8\times$–$2.8\times$ across all tasks. On Stack-Exchange with Qwen2.5-7B-Instruct, OPPO reaches a reward of 4.17 in 2,300 minutes versus 4,300 minutes for the baseline, yielding a $1.9\times$ speedup. With Qwen2.5-3B-Instruct on the same dataset, OPPO achieves a reward of 5.12 in 5,200 minutes compared to 13,000 minutes, corresponding to a $2.5\times$ improvement. These gains stem from two sources: (*i*) *intra-step overlap*, which hides reward prefilling latency during

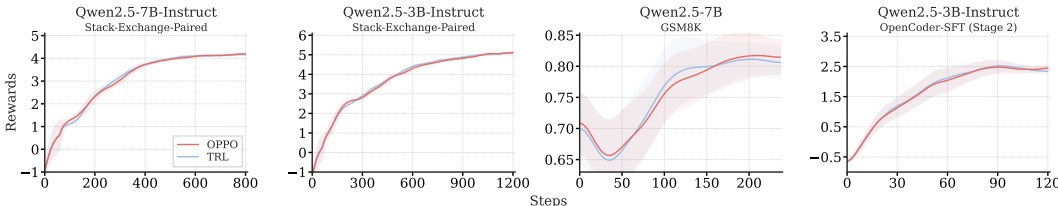

Figure 3: OPPO improves PPO-based RLHF training efficiency by $1.8\times$–$2.8\times$ over TRL across datasets, enabled by overlapping actor generation with reward scoring and early stopping.

Figure 4: OPPO achieves efficiency gains without affecting training quality.

actor decoding, and (*ii*) *inter-step overlap with dynamic* $\Delta$, which mitigates tail stragglers that would otherwise block shorter generations. OPPO achieves $2.4\times$ and $2.8\times$ speedup on OpenCoder-SFT (Stage 2) with Qwen2.5-3B-Instruct, and on GSM8K with Qwen2.5-7B, respectively.

**OPPO preserves training convergence.**   Despite substantial wall-clock speedups, Figure 4 shows that OPPO does not sacrifice training convergence. On Stack-Exchange, OPPO and the baseline follow nearly identical trajectories on both Qwen2.5-7B-Instruct and Qwen2.5-3B-Instruct training, such as reaching a reward of $\sim2.0$ by step 150, then plateauing at $\sim4.1$ by step 600 and $\sim5.12$ by step 1,000, respectively. On GSM8K with Qwen2.5-7B, both methods exhibit the same characteristic learning phases: an initial accuracy of 0.70, a dip to 0.66 around steps 25–50 as the model unlearns initial biases, and steady improvement to 0.82 by step 200. Finally, on OpenCoder-SFT (Stage 2) with Qwen2.5-3B-Instruct, both methods converge to a plateau around 2.4 by step 80. Across all tasks, the near-identical step-to-reward curves confirm that OPPO achieves a near-optimal balance between execution efficiency and convergence quality.

**OPPO largely boosts hardware resource utilization.**   Figure 5 shows that OPPO substantially improves GPU utilization. On the Stack-Exchange-Paired dataset with the Qwen2.5-7B-Instruct model, utilization increases from 50.6% to 71.0%, a $1.4\times$ improvement. With the Qwen2.5-3B-Instruct model on the same dataset, utilization rises from 38.7% to 73.6%, a $1.9\times$ improvement. On GSM8K with the Qwen2.5-7B model, OPPO boosts utilization from 45.7% to 67.3%, a $1.5\times$ improvement. On OpenCoder-SFT (Stage 2) with the Qwen2.5-3B-Instruct model, GPU utilization improves from 35.7% to 74.1%, corresponding to a $2.1\times$ increase. Note that utilization does not reach 100% because of unavoidable parallelism bubbles, memory stalls, and communication overheads.

**OPPO improves performance in multi-node settings.**   Table 1 shows that OPPO achieves $4.49\times$ reduction end-to-end step latency than TRL on Stack-Exchange-Paired with the Qwen2.5-7B-Instruct model across two nodes (each $4\times$A100-40GB).

**OPPO delivers improvements over different model parallelism plans.**   The distinct system-level benefits of OPPO are evaluated by comparing it against state-of-the-art frameworks, including VeRL (configured with data parallelism (DP), sequence parallelism (SP), and fully async w/ SP) and AReaL. Table 4 shows OPPO achieves the lowest latency (99.84s), outperforming VeRL w/ DP by $1.26\times$ and surpassing highly optimized systems such as AReaL and VeRL variants. These results suggest that OPPO targets a latency source distinct from sequence-level optimizations. While frameworks like VeRL and AReaL process responses only after full generation and leave the reward model idle, OPPO 's intra-step overlap streams intermediate chunks to utilize this time. Consequently, OPPO

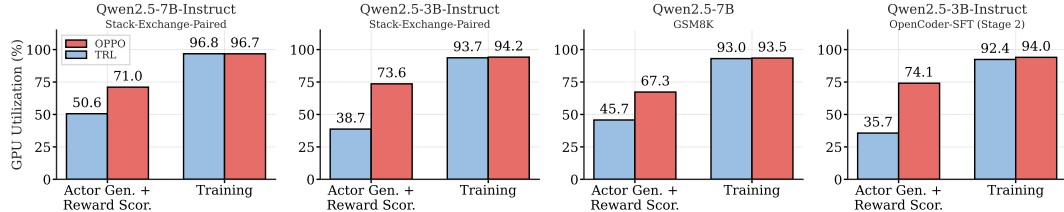

Figure 5: OPPO improves GPU utilization in the inference stage by $1.4\times$–$2.1\times$, enabling more efficient compute use by overlapping actor generation with reward scoring.

|                   | TRL    | OPPO   |
| ----------------- | ------ | ------ |
| Mean latency (s)  | 498.30 | 111.08 |
| Speed up          | 1.00x  | 4.49x  |

Table 1: OPPO achieves lower end-to-end step latency than TRL by $4.5\times$ in multi-node settings.

addresses a bottleneck orthogonal to sequence parallelism, making it a complementary optimization composable with existing strategies.

## 4.3 ABLATION STUDIES

**Performance Breakdown by Design Components.** We ablate OPPO into two variants to isolate the impact of each design choice: (1) *OPPO w/o Intra*, which disables intra-step overlap (i.e., streaming upstream decoding to the reward model), and (2) *OPPO w/o Inter*, which disables inter-step overlap (i.e., batch overcommitment with dynamic $\Delta$).

Figure 6 reports their performance on Stack-Exchange-Paired. For Qwen2.5-7B-Instruct, the TRL baseline requires 4,200 minutes to reach a reward of 4.17. Adding only intra-step overlap reduces this to 3,500 minutes ($1.2\times$ speedup), as streaming hides about 17% of scoring latency during generation. However, the gain is bounded by stragglers from the longest sequences in each batch. Applying only inter-step overlap reduces training time further to 2,700 minutes ($1.6\times$ speedup). For Qwen2.5-3B-Instruct, the TRL baseline requires 13,000 minutes to reach a reward of 5.12. Intra-step overlap reduces this to 10,000 minutes ($1.3\times$ speedup), while inter-step overlap achieves 6,300 minutes ($2.06\times$ speedup). Again, all configurations converge to similar final rewards, confirming that intra- and inter-step overlaps address orthogonal bottlenecks while preserving training quality.

**Robustness and Staleness.** As detailed in Algorithm 1, the $\Delta$ controller adapts to a windowed reward trend, updating $\Delta$ only through bounded, gradual steps. This design prevents abrupt jumps and effectively filters short-term oscillations. The request-deferral distribution in Table 2 confirms the stability of this approach: the vast majority of requests are processed immediately, and nearly all deferred requests are delayed by only a single step. This indicates neither perpetual deferral of difficult prompts nor excessive staleness that would affect rewards.

**Effectiveness of Inter-step Adaptation.** Figure 7a compares OPPO with fixed and dynamic $\Delta$. With fixed $\Delta = 4$, training converges more slowly since fewer long-tail generations are stopped early, limiting overlap benefits. Fixed $\Delta = 8$ accelerates convergence by skipping more long-tail generations, but its static threshold cannot adapt well across all phases of training. In contrast, dynamic $\Delta$

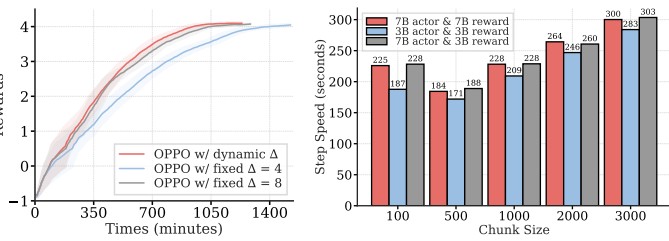

(a) Inter-step adaptation ($\Delta$).      (b) Impact of chunk size.

Figure 7: Ablation studies on efficiency: (a) fixed vs. dynamic $\Delta$, and (b) chunk size effect on step speed.

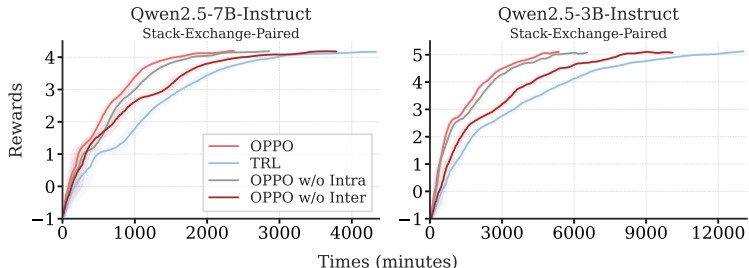

Figure 6: Performance breakdown showing the impact of OPPO's intra- and inter-step overlaps. Both optimizations drive the $1.8\times$–$2.8\times$ speedup without harming convergence quality.

| Deferred steps | 0 | 1 | 2 | 3 | Avg. deferred steps |
|---|---|---|---|---|---|
| Share of requests | 78.48% | 20.20% | 0.23% | 1.05% | 0.24 |

Table 2: Distribution of requests deferral shows most requests are not deferred, and nearly all others are delayed by only a single step.

consistently achieves the best performance by adapting the threshold over time, leading to faster convergence and more stable rewards. These results underscore that static choices of $\Delta$ create efficiency–stability tradeoffs, whereas dynamic $\Delta$ eliminates this tension by adjusting to the evolving distribution of rollout lengths throughout training.

**Impact of Streaming Degrees.**    Figure 7b shows the effect of chunk size on step speed across different model sizes. Small chunks (100 tokens) incur high scheduling and GPU context-switch overhead, reducing throughput despite high overlap. Moderate chunks (500 tokens) strike the best balance, yielding the fastest step speeds by maximizing overlap while avoiding overhead. Large chunks (1000–3000 tokens) reduce overlap opportunities and push execution closer to sequential mode, causing step times to rise again. These results highlight that throughput is highly sensitive to chunk size, and the optimal setting depends on model scale and workload.

**OPPO Preserves Final Accuracy.**    Table 3 shows that for the 3B model, OPPO consistently outperforms the TRL baseline across all benchmarks, with gains ranging from 0.07 to 0.92 percentage points (mean: 0.48 pp). For the 7B model, the differences are minimal (-0.24 to +0.25 pp; mean: +0.02 pp): OPPO achieves higher accuracy on ARC-Challenge, HellaSwag, and GSM8K, while showing slight declines on ARC-Easy and TruthfulQA-MC2. These fluctuations fall within the expected statistical variance of RLHF training. Overall, the comparable performance across both model scales confirms that OPPO's pipeline-overlap strategy accelerates training without sacrificing model quality.

**Applicability beyond PPO.**    OPPO's benefits extend to any online preference-optimization method involving variable-length on-policy generations (e.g., DPO or GRPO). These methods can adopt the same scheduling logic: generate $B+\Delta$ items, update on the first $B$ completions, and carry unfinished long generations forward to the next iteration. This strategy reduces tail latency without altering the optimization objective or the distribution of responses used for updates. As shown in Figure 3, OPPO achieves $2.35\times$ faster convergence in a rule-based PPO setting on GSM8K (without a reward model), confirming that the inter-step overlap mechanism remains effective even in non-standard or simplified RLHF pipelines.

## 5  RELATED WORK

**PPO-based RLHF Efficiency.**    Hydra-PPO (Santacroce et al., 2023) reduces memory and latency by combining LoRA with parameter sharing across actor, critic, and reward models. Offline PPO methods (Hu et al., 2023; Noukhovitch et al., 2025) improve stability and efficiency by training from fixed preference datasets, thereby avoiding costly online rollouts. Data-centric approaches such as LIMO (Ye et al., 2025) and S1 (Muennighoff et al., 2025) demonstrate that small, curated datasets

| Tasks | Qwen2.5-3B Model | | | Qwen2.5-7B Model | | |
|---|---|---|---|---|---|---|
| | TRL | OPPO | Change | TRL | OPPO | Change |
| ARC-Challenge | 48.89 | **49.57** | +0.68 | 55.55 | **55.80** | +0.25 |
| ARC-Easy | 74.54 | **75.08** | +0.54 | **81.57** | 81.36 | -0.21 |
| HellaSwag | 75.01 | **75.19** | +0.18 | 80.70 | **80.79** | +0.09 |
| TruthfulQA MC2 | 59.07 | **59.99** | +0.92 | **64.27** | 64.03 | -0.24 |
| GSM8K | 63.46 | **63.53** | +0.07 | 82.56 | **82.79** | +0.23 |
| **Average** | 64.19 | 64.67 | **+0.48** | 72.93 | 72.95 | **+0.02** |

Table 3: Evaluation results on core tasks (0-shot) and math tasks (5-shot). We report accuracy (%) for TRL-trained models and OPPO-trained models, along with the absolute change.

| | VeRL w/ DP | VeRL w/ DP+SP | AReaL | OPPO |
|---|---|---|---|---|
| Mean latency (s) | 125.36 | 120.47 | 109.92 | 99.84 |

Table 4: OPPO achieves the lowest per-step latency under identical hardware and rollout settings, suggesting system-level benefits beyond VeRL (DP, DP+SP, Fully Async w/ SP) and AReaL.

can yield competitive performance. LIMR (Li et al., 2025) prioritizes samples using impact-based scoring, while ADARFT (Shi et al., 2025) adopts a lightweight curriculum that adjusts difficulty through reward signals. These methods primarily optimize data or optimization strategy, whereas our work focuses on improving system-level efficiency by restructuring PPO's execution pipeline.

**Model Training Efficiency.** System-level techniques seek to accelerate RLHF training by rethinking the execution stack. TRL (von Werra et al., 2020a) provides scalable multi-node training with parameter-efficient fine-tuning. OpenRLHF (Hu et al., 2025b) integrates vLLM (Kwon et al., 2023) with Ray (Liaw et al., 2018) to accelerate generation and scheduling. HybridFlow (Sheng et al., 2025) improves throughput by combining single- and multi-controller paradigms, while RLHFuse (Zhong et al., 2025) boosts GPU utilization through stage fusion and micro-batch scheduling. Our approach complements these efforts by compounding PPO's disaggregated stages with intra- and inter-step overlap, further improving utilization and throughput.

**RLHF Optimizations.** Another active direction reduces algorithmic complexity or improves robustness. Critic-free algorithms—such as GRPO (Shao et al., 2024), ReMax (Li et al., 2024), RLOO (Ahmadian et al., 2024), and REINFORCE++ (Hu et al., 2025a)—remove the value network, estimating advantages directly from normalized rewards over multiple rollouts. RL-free methods including DPO (Rafailov et al., 2024) and EXO (Ji et al., 2024) bypass reinforcement learning entirely, while robustness-focused methods like RLP (Lang et al., 2024) and BSPO (Dai et al., 2025) mitigate reward misalignment. Other efforts, such as LoCo-RLHF (Lee et al., 2024), address preference heterogeneity. Our method is orthogonal to these algorithmic improvements, as it preserves PPO semantics while accelerating its execution.

## 6 CONCLUSION

We introduce OPPO, a lightweight framework for efficient PPO-based RLHF training by maximizing execution overlap. OPPO introduces a new dimension of efficiency—*intra-step overlap*, which streams actor tokens to downstream models for incremental prefilling, and *inter-step overlap*, which strategically defers stragglers to future steps. Both overlaps convert idle time into useful work. Our extensive evaluations across free-form generation, math reasoning, and code generation tasks, show that OPPO accelerates PPO training by up to $2.8\times$, raises GPU utilization by over $2.1\times$, and generalizes to alternative paradigms such as DPO.

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

# A  APPENDIX

## A.1  MODEL EVALUATION PROTOCOL

We conduct comprehensive evaluations to assess the impact of OPPO on model quality across different model scales. Our evaluation protocol compares models trained by TRL with standard PPO-based RLHF against models trained with our proposed overlapping optimization techniques. We evaluate two model configurations: Qwen2.5-7B and Qwen2.5-3B each fine-tuned on the Stack-Exchange-Paired dataset for comparable training steps.

### A.1.1  BENCHMARK SUITE

To ensure a comprehensive assessment of model capabilities, we employ the Language Model Evaluation Harness (Gao et al., 2024),a standardized framework for evaluating language models across diverse tasks. Our evaluation suite comprises six core benchmarks that assess different aspects of model performance:

**Reasoning and Common Sense Tasks:**

- **HellaSwag** (Zellers et al., 2019): Evaluates commonsense reasoning through sentence completion, requiring models to select plausible continuations of everyday scenarios.

- **ARC (AI2 Reasoning Challenge)** (Clark et al., 2018): Comprises two subsets—ARC-Easy and ARC-Challenge—assessing scientific reasoning through grade-school science questions of varying difficulty.

**Truthfulness and Mathematical Reasoning:**

- **TruthfulQA-MC2** (Lin et al., 2022): Measures the model's tendency to generate truthful responses through multiple-choice questions designed to elicit common misconceptions.

- **GSM8K** (Cobbe et al., 2021): Evaluates mathematical reasoning through grade school math word problems requiring multi-step solutions.

### A.1.2  EVALUATION METRICS

For each benchmark, we report multiple metrics to capture nuanced performance differences:

- **Standard Accuracy (acc)**: Raw accuracy scores computed directly from model predictions.

- **Normalized Accuracy (acc_norm)**: Length-normalized accuracy accounting for varying response lengths, particularly relevant for multiple-choice tasks.

- **Exact Match Scores**: For GSM8K, we report both strict-match scores (requiring exact numerical answers) and flexible-extract scores (allowing for minor formatting variations).

### A.1.3  EVALUATION PIPELINE

Our evaluation pipeline follows a systematic approach to ensure reproducible and reliable results:
**Stage 1: Environment Configuration.** Each evaluation begins with proper environment initialization, including CUDA device allocation and verification of GPU availability. We employ float16 precision for all evaluations to maintain consistency with training configurations while optimizing memory utilization.

**Stage 2: Batch Processing.** Models are evaluated using adaptive batch sizing based on available GPU memory. For 7B models, we utilize a batch size of 4, while 3B models support a batch size of 8, maximizing throughput without encountering out-of-memory errors. All evaluations employ greedy decoding to ensure deterministic and reproducible results.

**Stage 3: Task-Specific Evaluation.** Each benchmark task is evaluated independently to isolate performance characteristics. The evaluation harness automatically handles task-specific preprocessing, including few-shot prompt construction where applicable. For reasoning asks (ARC, HellaSwag), we employ 25-shot, 10-shot, and 5-shot evaluations, respectively, following established protocols.

TruthfulQA-MC2 uses 0-shot evaluation to assess inherent model knowledge without exemplar influence.

### A.1.4 STATISTICAL CONSIDERATIONS

To ensure statistical validity of our comparisons, we maintain consistent evaluation conditions across all model pairs:

- Fixed random seeds for reproducible prompt sampling
- Identical prompt formulations and few-shot examples
- Consistent tokenization and preprocessing pipelines
- Synchronized evaluation checkpoints (e.g., 800 steps for 7B models, 1200 steps for 3B models)

Evaluations are conducted on NVIDIA A100 40GB GPUs, with a complete evaluation of a single model requiring approximately 2-3 hours depending on model size. The evaluation pipeline is designed to be portable and reproducible, with automated dependency management and environment configuration scripts to facilitate replication across different computational environments.

### A.2 THE USE OF LARGE LANGUAGE MODELS (LLMS)

We used a large language model (ChatGPT) only as a writing assistant tool to check grammar, improve readability, and polish sentence clarity.

### A.3 REPRODUCIBILITY STATEMENT

To ensure reproducibility of our results, we provide comprehensive implementation details throughout the paper and supplementary materials. The complete OPPO algorithm is specified in Algorithm 1, with detailed descriptions of the intra-step and inter-step overlap mechanisms in Sections 3.1 and 3.2. Experimental configurations, including model architectures and training settings for both 3B and 7B models, are detailed in Section 4.1 and Appendix A. We utilize publicly available datasets with standard preprocessing procedures described in Section 4. The dynamic control parameters for overcommitment degree adaptation are fully specified in Sections 3.1 and 3.2, including bounds, momentum values, and window sizes. Our implementation requires minimal modifications to existing PPO codebases, with the specific integration points outlined in Section 3. All experiments were conducted on NVIDIA A100 and H200 GPUs. We will release our implementation code upon publication to facilitate reproduction and adoption of OPPO in existing RLHF pipelines.

