# OpenReview forum: "OPPO: Accelerating PPO-based RLHF via Pipeline Overlap"
_ICLR.cc/2026/Conference — ICLR 2026 Poster_

### Official Review · Reviewer_FKvv · 2025-10-29

**Soundness:** 2
**Presentation:** 2
**Contribution:** 2
**Rating:** 4
**Confidence:** 4

**Summary:**

This paper examines training system efficiency in PPO-based RLHF algorithms, revealing significant GPU underutilization and latency bottlenecks in current synchronous approaches. To overcome these limitations, the authors introduce a dual-strategy solution: First, intra-step overlap enables pipelined execution by chunking upstream model outputs (e.g., actor model) and streaming them to downstream models (e.g., reward model), allowing prefill operations to commence before upstream decoding completes. Second, inter-step overlap employs adaptive prompt overcommitment, strategically postponing lengthy generations to subsequent steps to reduce tail latency while preserving computational progress. Experimental results on the TRL framework demonstrate 1.8×–2.8× efficiency gains over baseline implementations.

**Strengths:**

- Provides valuable empirical analysis of inefficiencies in practical PPO-based training systems.
- Introduces effective actor generation and reward scoring overlap strategies to improve pipeline efficiency.
- Demonstrates promising empirical results on small models, achieving reduced running time without performance degradation.

**Weaknesses:**

- The paper lacks systematic analysis of system efficiency. A more principled approach to analyzing the dimensions for reducing time complexity and examining related hyperparameters would strengthen the contribution.

- The improvements are demonstrated on small-scale models with limited training data. It remains unclear whether these advantages extend to large-scale models or architectures beyond those tested in the paper. A theoretical framework and analysis would make the claims more convincing.

- The paper lacks comparison with existing systems such as Verl, OpenRLHF, AReaL, and RoLL. These systems have explored numerous resource management and load-balancing strategies for addressing challenges like long-tail generation.

- The improvements are restricted to PPO algorithms in the RLHF setting, which limits the paper's impact. A significant portion of modern alignment training uses GRPO and RLVR approaches, where no reward model is involved, and the proposed overlap strategies may not be applicable.

**Questions:**

- Regarding the intra-step overlap strategy, could this be implemented entirely at the sequence level? Specifically, would it be feasible to use an asynchronous approach where the generation batch is split into mini-batches, with each completed sequence immediately forwarded to the reward model for scoring upon generation completion?
- Could the authors provide a detailed comparison with recent advances in Verl, AReaL, RoLL, and OpenRLHF? These represent more modern, industry-grade systems with their own optimization strategies. A comparison would help clarify the unique contributions of this work relative to the current state-of-the-art, as TRL may not reflect the most competitive baseline.

---

> ### Author Response · Authors · 2025-11-24
>
> We thank you for the insightful feedback.
>
> ***Q1: Regarding the intra-step overlap strategy, could this be implemented entirely at the sequence level? Would it be feasible to use an asynchronous approach?***
>
> We note that sequence-level asynchronous optimization is complementary to OPPO design: sequence-level parallelism processes the response in parallel ***after the entire sequence (actor response) finishes***, so the reward model still remains idle while the actor is decoding long responses. OPPO’s intra-step overlap targets this bottleneck directly by streaming intermediate chunks, enabling the reward/value models to begin prefilling while decoding is still ongoing without alternating the generated response (i.e., lossless). This improves GPU utilization, particularly for long-tail generations, without altering the PPO objective.
>
> To further validate that OPPO adds benefits beyond sequence-level asynchrony, we added new experiments comparing against VeRL fully-async PPO with sequence parallelism(SP).
>
> | Metric (s/step) | VeRL Fully-Async PPO with SP | OPPO |
> | :--- | :--- | :--- |
> | Mean latency | 161.21 | 99.84 |
>
> Overall, these results show that OPPO preserves convergence under a strong fully-async baseline.
>
> ***Q2: Detailed comparison with recent advances in VeRL, AReaL, RoLL, and OpenRLHF?***
>
> We note that OPPO is complementary to these systems as discussed in Q1. We further compared OPPO directly with VeRL and AReaL under the same workload and resource settings. The goal of this comparison is to isolate whether OPPO’s overlap mechanisms provide benefits beyond what these optimized stacks already achieve.
>
> A compact view of the steady-state benefit is:
> | Metric (s/step) | VeRL | AReaL | OPPO |
> | :--- | :--- | :--- | :--- |
> | Mean latency | 125.36 | 109.92 | 99.84 |
> | Speedup vs VeRL | 1.00x | 1.14x | 1.26x |
>
> These numbers show that while VeRL and AreaL are already efficient, OPPO’s overlap design still provides an additional system-level gain.

---

### Official Review · Reviewer_ty1D · 2025-11-01

**Soundness:** 3
**Presentation:** 3
**Contribution:** 3
**Rating:** 6
**Confidence:** 3

**Summary:**

The paper focuses on improving the training efficiency of PPO algorithms. The authors propose two overlapping techniques:
- Intra-step overlap: allows the downstream model (e.g., reward model) to begin prefill while the upstream model continues decoding.
- Inter-step overlap: adaptively overcommits a few prompts and defers longer generations to future steps.
Experiments demonstrate that the proposed methods significantly accelerate PPO-based RLHF training.

**Strengths:**

- The paper is well written and easy to follow.

- Training efficiency of PPO algorithms is an important problem. The proposed two overlapping techniques are intuitive and well motivated.

- Experiments across different tasks are conducted to demonstrate the acceleration achieved by the proposed algorithm.

**Weaknesses:**

- For intra-step overlap, in reasoning scenarios, difficult problems often require generating long responses. With this technique, is there a risk that updates for difficult problems will always be deferred, while the algorithm converges quickly on easier problems?

- Could intra-step overlap affect the distribution being learned, as discussed in the first question?

- There are other techniques, such as fine-grained parallelism adopted in VERL for accelerating PPO, but these are not included in the experiments. Could the authors elaborate on the reason for this omission?

**Questions:**

Please refer to the weakness part.

---

> ### Author Response · Authors · 2025-11-24
>
> We thank you for the insightful feedback.
>
> ***Q1 and Q2: Is there a risk that intra-step overlap will always defer some requests, while the algorithm converges quickly on easier problems? Could intra-step overlap affect the distribution being learned?***
>
> We kindly note that we have proved that intra-step overlap does not alter the response generation (Section 3.1)—it is a pure systems optimization that performs incremental prefilling to overlap the decoding stage of upstream response generation and downstream reward prefilling—thus not affecting the distribution being learned. Our performance breakdown (Figure 4) further shows that intra-step overlap does not change the step-to-reward curve.
>
> The deferral in inter-step overlap with OPPO design is marginal: (1) in each step, OPPO updates on the first $B$ completed rollouts, while the remaining $\Delta$ unfinished long generations are moved to the head of the generation request buffer, thus having higher priority than any new requests, and continue decoding in the next step; and (2) OPPO automatically adjusts the $\Delta$ value to preserve convergence. Both designs explicitly prevent starvation of difficult prompts; they are delayed briefly but not discarded, and partial work is preserved. Our evaluations across diverse models and benchmarks show that inter-overlap does not hurt final model performance while achieving substantial efficiency improvements.
>
> To further quantify the magnitude of deferral in a reasoning-style setting, we added new StackExchange experiments with the same configuration as in the main paper. We observed that deferral is small and tightly bound: the maximum deferral is 3 steps (~1% requests), and most deferred requests are delayed by only 1 step. As the table below shows, the average deferred steps per request is 0.24, meaning the vast majority of updates still include difficult prompts after at most a very short delay.
>
> | Deferred steps | 0 | 1 | 2 | 3 | Avg. deferred steps |
> | :--- | :--- | :--- | :--- | :--- | :--- |
> | Share of requests | 78.48% | 20.20% | 0.23% | 1.05% | 0.24 |
>
>
> ***Q3: There are other techniques, such as fine-grained parallelism, adopted in VERL for accelerating PPO, but these are not included in the experiments?***
>
> VeRL’s fine-grained parallelism is an orthogonal systems optimization that focuses on model placement and fine-grained task scheduling across GPUs. We did not include it in the original evaluation because enabling it introduces additional synchronization/coordination overheads and several interacting tuning knobs, which would make it difficult to isolate the effect of OPPO’s overlap mechanisms in a controlled ablation. Importantly, OPPO is designed to be complementary to such mapping strategies rather than a replacement.
>
> To directly address this concern, we added new experiments under identical resources comparing (1) VeRL with data parallelism (DP), (2) VeRL with DP + sequence parallelism (SP), and (3) OPPO. The averaged per-step latencies are:
>
> | Metric (s/step) | VeRL-DP | VeRL-DP + SP | OPPO |
> | :--- | :--- | :--- | :--- |
> | Mean latency | 125.36 | 120.47 | 99.84 |
>
> These results show that OPPO outperforms even the very state-of-the-art PPO designs like VeRL sequence parallelism by 1.3x, and we will include this comparison in the final version.

---

### Official Review · Reviewer_Hdb3 · 2025-11-02

**Soundness:** 3
**Presentation:** 3
**Contribution:** 3
**Rating:** 8
**Confidence:** 3

**Summary:**

This paper identifies and addresses two major sources of inefficiency in standard PPO-based RLHF training pipelines: (1) sequential dependencies between the actor, critic, and reward models, where downstream models must wait for the upstream actor to fully complete generation , and (2) long-tail latency, where a few long "straggler" responses in a batch delay the completion of the entire step .

The authors propose OPPO, a lightweight systems optimization framework that introduces two novel overlap techniques:

Intra-step Overlap: The actor model streams its generated tokens in adaptive chunks to the downstream critic and reward models. This allows the downstream models to begin their computationally expensive "prefill" stage concurrently while the actor is still decoding, effectively hiding the scoring latency .

Inter-step Overlap: The pipeline "overcommits" by starting with a batch of B + Δ prompts. It sends the first B responses that complete to the training stage, while the Δ unfinished straggler responses are deferred and resumed in the next iteration. This mitigates tail latency without discarding partial work .

Empirically, OPPO is shown to accelerate end-to-end PPO training by 1.8×–2.8× and improve GPU utilization by 1.4×–2.1×, all while achieving identical step-to-reward convergence and final model quality

**Strengths:**

Massive Efficiency Gains: The headline result of a 1.8×–2.8× speedup on a notoriously slow and expensive pipeline is a major strength. This is a huge practical win.

Preservation of Convergence: The paper's strongest evidence is that this speedup comes "for free." Figure 4 and Table 1 show that the step-to-reward convergence and final model quality are nearly identical to the baseline. This proves it's a true systems optimization, not an algorithmic trade-off.

Orthogonal Solution: The method attacks two different, orthogonal bottlenecks: pipeline bubbles (via intra-step overlap) and straggler latency (via inter-step overlap). The ablation in Figure 6 clearly shows that both components are necessary for the full speedup.

Smart Adaptive Control: The heuristic for dynamically controlling the overcommitment level Δ is very clever. It adaptively becomes more aggressive when training is improving and more conservative (decaying Δ) as training converges, balancing speed with stability.

**Weaknesses:**

"Lightweight" Claim: The paper claims the method is "lightweight" and requires "a few lines of code change". This seems like a significant overstatement. Implementing a streaming data buffer, inter-model communication for chunks, and the dynamic Δ scheduler (Algorithm 1)  is a non-trivial systems engineering task that deeply modifies the training loop's data flow.

Heuristic-Driven Controls: The method's dynamic controls are entirely heuristic. The chunk size is tuned by "periodically... appl[ying] a few candidate chunk sizes" , and the Δ level is based on the reward slope. While these heuristics are smart and work well, they are not theoretically derived and may require careful tuning (e.g., the window size W, bounds Δmin/Δmax).

Unanalyzed Staleness: The inter-step overlap introduces a form of staleness. While the policy used for generation is consistent (the work is just deferred), the prompts that are deferred are from an older data distribution. The paper shows that high staleness is bad (Fig 2c)  but doesn't analyze if this prompt-distribution staleness has any subtle, negative effects on the converged model, especially if Δ is large.

**Questions:**

Could you elaborate on the "few lines of code change" claim? This appears to require a complex streaming and scheduling data structure (Algorithm 1) . What is the practical engineering lift to integrate OPPO into an existing TRL-based pipeline?

The dynamic control for Δ is based on the reward slope st. How sensitive is this heuristic to a noisy reward signal? If the reward oscillates (common in RL), would this cause Δ to become unstable and harm performance?

The paper mentions OPPO "generalizes to other paradigms such as DPO". How would this work? The DPO pipeline does not have the same multi-stage, multi-model dependency as PPO. What exactly would be overlapped?

---

> ### Author Response · Authors · 2025-11-24
>
> We thank you for the insightful feedback.
>
> ***Q1. Could you elaborate on the "few lines of code change" claim? This appears to require a complex streaming and scheduling data structure (Algorithm 1). What is the practical engineering lift to integrate OPPO into an existing TRL-based pipeline?***
>
> Herein, “a few lines of code change” implies the changes to the user-facing training script or config, not to the internal implementation of OPPO itself. In practice, OPPO is implemented as a small runtime module that wraps an existing PPO trainer. The OPPO runtime encapsulates the streaming and scheduling logic in Algorithm 1; integrating it into an existing TRL-based pipeline only requires:
> 1. Replacing the standard PPO trainer call with an OPPO trainer wrapper, and
> 2. Adding a few configuration fields (e.g., check size and initial delta number in the config file.
>
> In our implementations, this corresponds to fewer than 10 lines of change in the training script: no changes to the PPO loss, optimizer, or model architectures are needed. We will clarify this in our final version and open-source the implementation.
>
> ***Q2. The dynamic control for $\Delta$ is based on the reward slope st. How sensitive is this heuristic to a noisy reward signal? If the reward oscillates (common in RL), would this cause $\Delta$ to become unstable and harm performance?***
>
> Our dynamic $\Delta$ controller is intentionally conservative to make it robust to noisy rewards. Concretely, we compute the reward slope $s_t$ over a window of $W$ PPO steps (in our experiments $W=27$ steps). Only once every $W$ steps do we consider adjusting $\Delta$. At each adjustment:
> 1. $\Delta$  is changed by at most $\pm$ 2
> 2. We clamp delta to the range $[0, B/2]$; with batch size $B = 16$, the maximum $\Delta$ is 8
>
> Thus, $\Delta$ evolves slowly: it cannot jump abruptly, and short-term oscillations in the per-step reward are smoothed by the windowed slope and the small step size. Figure 7 shows that dynamic $\Delta$ yields faster convergence than any fixed $\Delta$ while the reward curve remains smooth and does not exhibit drastic instability; this is precisely because the controller reacts only to consistent trends, not to single noisy updates.
>
> To further show that deferral does not introduce harmful staleness, the table below demonstrates that the average deferred steps per request is only 0.24, with nearly all requests delayed by no more than a single step.
>
> | Deferred steps | 0 | 1 | 2 | 3 | Avg. deferred steps |
> | :--- | :--- | :--- | :--- | :--- | :--- |
> | Share of requests | 78.48% | 20.20% | 0.23% | 1.05% | 0.24 |
>
> ***Q3: The paper mentions OPPO "generalizes to other paradigms such as DPO". How would this work? The DPO pipeline does not have the same multi-stage, multi-model dependency as PPO. What exactly would be overlapped?***
>
>
> Although DPO and GRPO do not introduce expensive RM/critic dependencies, OPPO's inter-step overlap design can optimize their long, variable-length on-policy generation: generate $B+\Delta$ prompts, update on the first $B$ finished pairs, and carry long generations forward, hiding tail latency. Indeed, we evaluated OPPO under a rule-based PPO setting with no learned reward model. As shown in Figure 3, OPPO still achieves 2.35× faster convergence on GSM8K, confirming that inter-step overlap remains effective even without neural scoring. We will revise the paper to clarify this.

---

### Official Review · Reviewer_bNZ4 · 2025-11-03

**Soundness:** 3
**Presentation:** 3
**Contribution:** 3
**Rating:** 8
**Confidence:** 3

**Summary:**

This paper introduces a new pipeline methodology for PPO. The methodology is illustrated for single-node training and results in significantly faster training times.

**Strengths:**

The impacts on the speed of post training are quite significant, and the implementation appears to be easy to add to existing code bases. The paper is well-written with detailed experiments and clear plots. I also really appreciate the annotated algorithm.

**Weaknesses:**

> Among RLHF methods, Proximal Policy Optimization (PPO) (Schulman et al., 2017) has been the de facto standard due to its training stability and flexibility across diverse reward models and objectives. PPO underpins the training of state-of-the art LLMs such as GPT-4 (OpenAI et al., 2024), Gemma (Team et al., 2024), and LLaMA (Touvron et al., 2023; Grattafiori et al., 2024).

The quoted passage is wrong. As far as I am aware there is no public information on the RLHF strategies used in GPT-4 or Gemma 3, and while LLaMA 2 (Touvron et al., 2023) used PPO, LLaMA 3 and 4  (Grattafiori et al., 2024; Meta, 2025) use DPO. I do not believe that PPO is the current standard anymore, with both DPO and GRPO being significantly more popular. Qwen 2 used DPO, Qwen 2.5 used both DPO and GRPO, and Qwen 3 used just GRPO. GRPO is used by DeepSeek in all their models. DPO is used by Tulu, Hermes3, and several other models as well. I am unaware of any major model trained in 2024 or 2025 that discloses that it uses PPO. None of this means that improving PPO's efficiency isn't useful, but this paper heavily oversells how widely used PPO is and uses that as a major motivator.

More importantly, this paper appears to exclusively examine single-node training. While that's not inherently problematic it is a strong limitation to the impact of the methodology since it would be inapplicable for large models. In the absense of any discussion or results of multinode training, I'm going to assume that the method doesn't work in a multinode settting. I strongly recommend including this as a limitation in the discussion of the paper.

**Questions:**

Do you have any results in a multinode setting?

---

> ### Author Response · Authors · 2025-11-24
>
> We thank you for the insightful feedback.
>
> ***Q1. PPO Use for ChatGPT?***
>
> We agree that public disclosures about the exact post-training algorithms for some recent proprietary models (e.g., GPT-4, Gemma 3) are limited, and our original wording overstated what is publicly known. We will revise the passage to avoid attributing specific RLHF optimizers to models without explicit documentation.
>
> In addition, we want to kindly note that OPPO design can benefit other RLHF paradigms, including GRPO. This is because of the prevalence of long-tailed response generation, which motivates our inter-step optimization. Indeed, Our rule-based PPO experiment (Figure 3) uses a rule-based reward verifier instead of a reward model, which resembles the GRPO setting. It shows that OPPO still brings clear benefits that speed up convergence (2.35× faster on GSM8K with Qwen2.5-7B) and keeps the reward curve stable. This shows that OPPO is a general system-level improvement, not something that depends on a neural reward model.
>
> That said, PPO-based RLHF remains an important and widely used class of reward-model-based online alignment methods. PPO is the canonical and most studied form of online RLHF. The InstructGPT alignment pipeline is explicitly PPO-based, and this actor-critic style remains a core reference for much of the community [1]. Moreover, recent studies show that PPO is not obsolete. When tuned properly, it can match or outperform DPO on dialogue, reasoning, and code tasks [2]. This motivates continued work on improving PPO efficiency, especially in scenarios with richer or multi-objective rewards [3].
>
> Finally, PPO remains supported in scalable alignment toolchains for large models. For example, NeMo-Aligner includes distributed PPO-RLHF with 3D parallelism and is designed to run at thousand-GPU scale, showing continued practical relevance beyond small setups [4].
>
> [1] Ouyang et al., Training language models to follow instructions with human feedback, 2022. https://arxiv.org/abs/2203.02155
>
> [2] Zhao et al., PPO can match or outperform DPO when properly tuned, 2024. https://arxiv.org/abs/2404.10719
>
> [3] Jiang et al., Pairwise Proximal Policy Optimization: Large Language Model Alignment via Comparative RL, UCB, 2024.
>
> [4] NVIDIA, NeMo-Aligner: Scalable LLM Alignment with RLHF, 2024. https://arxiv.org/abs/2405.01481
>
> ***Q2. Do you have any results in a multinode setting?***
>
> We added additional multi-node experiments using two GPU nodes (each 4×A100-40GB), following our previous settings. The results below show that OPPO's efficiency gains persist and remain substantial.
>
> | Metric (s/step) | Baseline multinode PPO (TRL) | OPPO |
> | :--- | :--- | :--- |
> | Mean latency | 498.30 | 111.08 |
> | Speedup vs baseline | 1.00x | 4.49x |
>
> These results demonstrate that OPPO is not limited to single-node training; under multi-node resource partitioning, it delivers ~4× reduction in per-step latency.

---

### Author Response · Authors · 2025-11-28

We have incorporated all clarifications and updates from our rebuttal into the revised manuscript. We kindly invite the reviewers to take a look at the updated version and let us know if there are any remaining questions or points that would benefit from further discussion. We would greatly appreciate any updates to your assessments based on the revisions. Thank you again for your time and thoughtful feedback.

---

### Author Response · Authors · 2025-12-02
**Summary of Reviewer Feedback and Corresponding Updates**

Dear AC,

We would like to express our gratitude to you for your moderation and timely help during the review and rebuttal session, and to all the reviewers for their constructive feedback. In particular, we appreciate that the reviewers recognized several positive aspects of our method OPPO, including: “clear and interesting observations and findings” (All reviewers), “clear motivation and practical relevance” (bNZ4, Hdb3), “strong empirical improvements” (Hdb3, ty1D, FKvv), and “valuable suggestions that strengthened our comparisons and analysis” (ty1D, FKvv).

For the comments and concerns raised in the reviews, we have prepared individual, ***point-by-point responses for each reviewer to address all the questions***. We have also revised the manuscript accordingly, and we summarize the key changes below. To facilitate cross-referencing, all modifications are highlighted in blue in the updated version.

Overall, our new version includes substantial updates, featuring 3 new tables (Tables 1, 2, and 4) and expanded discussions in Sections 1, 4.2, and 4.3, reflecting the detailed clarifications, additional benchmarks, and robustness analyses inspired by the reviewers’ feedback.

## Update Summary:
***1. Additional evaluation and analysis:***
- $\color{blue}\text{Table 1 in Section 4.1}$: Multi-node performance evaluation comparing OPPO and TRL (suggested by Reviewer bNZ4)
- $\color{blue}\text{Table 2 in Section 4.3}$: Request-deferral distribution analysis (suggested by Reviewers Hdb3 and ty1D)
- $\color{blue}\text{Table 4 in Section 5}$: Performance comparison with state-of-the-art frameworks, including VeRL variants and AReaL (suggested by Reviewers ty1D and FKvv)

***2. Writing***:
- Updated Abstract and $\color{blue}\text{Section 1}$ to refine novelty and updated terminology to "a lightweight wrapper" (replacing "a few lines of code") to accurately describe ease of integration (suggested by Reviewer Hdb3)
- Updated $\color{blue}\text{Section 1}$ to clarify PPO significance and relevance (suggested by Reviewer bNZ4)
- Added new discussion in $\color{blue}\text{Section 4.2}$ to the analysis of multi-node scalability comparing OPPO and TRL (suggested by Reviewer bNZ4)
- Added new discussion in $\color{blue}\text{Section 4.2}$ to analysis against different model parallelism plans (e.g., VeRL variants, AReaL), highlighting that OPPO is complementary to existing strategies (suggested by Reviewers ty1D and FKvv)
- Added new discussion in $\color{blue}\text{Section 4.3}$ to clarify the dynamic $\Delta$ controller and analyzed deferral distribution to confirm stability (suggested by Reviewers Hdb3 and ty1D)
- Added new discussion in $\color{blue}\text{Section 4.3}$ to explain how OPPO’s benefits extend to any online preference-optimization method involving variable-length on-policy generations (e.g., DPO or GRPO) (suggested by Reviewer Hdb3)

Overall, we believe the additional evaluations, detailed clarifications, and further discussion have fully addressed the reviewers' concerns. We will ensure that all new results, evaluations, and clarifications will be included in the final version of the paper, and we hope this summary helps provide a clear overview of our contributions and responses.

Thanks again,

Authors of OPPO

---

### Meta-Review · Area_Chair_2eS3 · 2026-01-06

**Summary:**

Summary The paper proposes OPPO, a framework to accelerate PPO-based RLHF training by introducing two pipeline overlap techniques: intra-step overlap (streaming actor outputs to downstream models) and inter-step overlap (deferring long-tail generations to subsequent steps). The authors claim significant speedups (1.8x–2.8x) and improved GPU utilization without compromising convergence.

Reviewers generally appreciated the practical value of the system optimizations and the substantial efficiency gains demonstrated in the experiments. Initial concerns focused on the relevance of PPO in the current landscape (vs. DPO/GRPO), the lack of multi-node scaling evaluation, comparisons against state-of-the-art systems (e.g., VeRL), and potential bias introduced by deferring difficult prompts.

Given the comprehensive rebuttal which provided additional benchmarks against SOTA frameworks (VeRL, AReaL), multi-node results, and staleness analysis, the AC recommends acceptance. The paper offers a valuable engineering contribution to the efficiency of online RLHF training.

**Reviewer Concerns:**

Reviewer Concerns The authors have conducted a very thorough rebuttal that addressed the majority of the substantial concerns raised by the reviewers:

Comparison with State-of-the-Art Systems (Addressed): Reviewers FKvv and ty1D rightly pointed out the lack of comparison with modern frameworks like VeRL and AReaL. The authors provided new experiments showing OPPO achieves lower latency and higher speedups (e.g., 1.26x vs AReaL) compared to these baselines under identical resource settings. This effectively addressed the concern that the gains were only relative to a weak baseline (TRL).

Multi-node Scalability (Addressed): Reviewer bNZ4 noted the lack of multi-node experiments. The authors added results using 2 nodes (4x A100 each), demonstrating a ~4.5x speedup, confirming the method scales beyond single-node setups.

Prompt Deferral and Bias (Addressed): Reviewers Hdb3 and ty1D were concerned that deferring long generations might bias training against difficult prompts or introduce staleness. The authors provided a distribution analysis showing the average deferred steps is negligible (0.24 steps) and confirmed via ablation that this does not harm convergence or model quality.

Relevance of PPO (Addressed): Reviewer bNZ4 criticized the motivation based on claims that PPO is used in GPT-4/Llama-3. The authors acknowledged the inaccuracy regarding proprietary model details and agreed to revise the text. They also persuasively argued for PPO's continued relevance in complex reward scenarios and noted the applicability of the overlap techniques to other paradigms like GRPO.

Reviewer FKvv’s request for a deeper theoretical framework for system efficiency remains partially open, as the paper is primarily empirical. However, for a systems track paper, the strong empirical evidence is sufficient.

**Reviewer Scores:**

Reviewer Scores Based on the strength of the rebuttal and the additional experiments provided, I estimate the final reviewer stance as follows:

Reviewer bNZ4: Initial Score: 8. Final Score: 8. The reviewer was already positive. The addition of multi-node experiments and the correction of the motivation text regarding SOTA models solidify this score.

Reviewer Hdb3: Initial Score: 8. Final Score: 8. The reviewer's concerns about "staleness" and the "lightweight" claim were adequately defended by the data on deferred step distribution and API explanation.

Reviewer ty1D: Initial Score: 6. Final Score: 7 or 8. The lack of comparison with VeRL was the main blocker preventing a higher score. Since the authors provided a direct comparison showing OPPO's superiority, this reviewer would likely upgrade their assessment significantly.

Reviewer FKvv: Initial Score: 4. Final Score: 5 or 6. This reviewer was the most critical regarding baselines (VeRL, OpenRLHF, AReaL). The authors explicitly addressed this by providing the requested comparisons in the rebuttal. While the reviewer also asked for theoretical analysis, the empirical superiority over the requested baselines should be sufficient to flip the recommendation to acceptance.

---

### Decision · Program_Chairs · 2026-01-26

Accept (Poster)